# Method to Solve Underwater Laser Weak Waves and Superimposed Waves

**DOI:** 10.3390/s23136058

**Published:** 2023-06-30

**Authors:** Chuanli Kang, Zitao Lin, Siyi Wu, Jiale Yang, Siyao Zhang, Sai Zhang, Xuanhao Li

**Affiliations:** 1College of Geomatics and Geoinformation, Guilin University of Technology, Guilin 541004, China; 2014012@glut.edu.cn (C.K.); 2120222033@glut.edu.cn (S.W.); 2120201716@glut.edu.cn (J.Y.); 2120211915@glut.edu.cn (S.Z.); zhangsai@glut.edu.cn (S.Z.); 2120221996@glut.edu.cn (X.L.); 2Key Laboratory of Spatial Information and Geomatics, Guilin University of Technology, Guilin 541004, China

**Keywords:** laser radar, waveform solution, LM algorithm, underwater exploration

## Abstract

With the rapid development of Lidar technology, the use of Lidar for underwater terrain detection has become feasible. There is still a challenge in the process of signal resolution: the underwater laser echo signal is different to propagating in the air, and it is easy to produce weak waves and superimposed waves. However, existing waveform decomposition methods are not effective in processing these waveform signals, and the underwater waveform signal cannot be correctly decomposed, resulting in subsequent data-processing errors. To address these issues, this study used a drone equipped with a 532 nm laser to detect a pond as the study background. This paper proposes an improved inflection point selection decomposition method to estimate the parameter. By comparing it with other decomposition methods, we found that the RMSE is 2.544 and R^2^ is 0.995975, which is more stable and accurate. After estimating the parameters, this study used oscillating particle swarm optimization (OPSO) and the Levenberg–Marquardt algorithm (LM) to optimize the estimated parameters; the final results show that the method in this paper is closer to the original waveform. In order to verify the processing effect of the method on complex waveform, this paper decomposes and optimizes the simulated complex waveforms; the final RMSE is 0.0016, R^2^ is 1, and the Gaussian component after decomposition can fully represent the original waveform. This method is better than other decomposition methods in complex waveform decomposition, especially regarding weak waves and superimposed waves.

## 1. Introduction

In recent years, deep-sea exploration has attracted considerable attention because of its usefulness in resource availability, defense, and transportation [1]. With the concept of Digital Earth proposed by the academician Guo [2], modern technologies have been widely used in the surveying and mapping industry, and the technology of underwater detection tends to be rich and diversified.

Several scholars have conducted research on the detection of underwater resources using different sensors. Huo et al. proposed a binocular vision-based underwater target detection and 3D reconstruction system [3]. Chen et al. used sonar techniques to finish damage detection in the underwater foundations of a bridge [4]. Shen et al. proposed a chaotic-pulse laser with high frequency intensity modulation for underwater target detection [5]. Nadimi applied synthetic aperture sonar to an underwater gas pipeline, and proposed a new method to improve detection efficiency [6]. With the differences in technologies and devices, the commonly used underwater detection methods are: laser radar, binocular vision camera, and sound radar. The underwater environment usually has uneven illumination, low visibility, and complexity. Sound radar has the widest application range, but it is vulnerable to interference in the complex and changeable underwater environment. The binocular vision camera has a high degree of visualization; however, compared with the other two detection methods, its accuracy is not very high, and it is easily affected by brightness and water quality. Laser radar is susceptible to the water quality and, with the increasing maturity of Lidar technology, the penetration of the laser has also been enhanced to a certain extent. Zou et al. implemented a hybrid Lidar for imaging remote objects underwater [7]. Li et al. found that a 532 nm laser with 500 MHZ could emit a stable and powerful modulated signal, which greatly increased the range of underwater detection, and ultimately obtained clear 3D and 4D images of the target underwater, across 20 m [8]. Li et al. found, through experiments, that the ranging error is approximately 12 cm in a 3 m water tank, using a 532 nm laser [9]. Menna, Agrafiotis, and Georgopoulos proposed airborne Lidar as a promising and effective technology for coastline topography and bathymetry [10]; however, the accuracy of underwater measurement is still not enough.

In recent years, some scholars have encountered this deficiency, and have proposed methods for underwater echo decomposition in shallow-water cases. Yang et al. proposed an airborne Lidar bathymetric waveform decomposition method, and achieved a good effect [11]. Liu et al. proposed an underwater decomposition and feature extraction based on airborne Lidar; it is considered that this can provide a basis for research on coastline separation, complex shallow-water-area sounding, water and underwater integrated splicing, and seabed classification, etc. [12]. Schwarz deeply analyzed the process of using Lidar to measure underwater depth, and highlighted the troubles and importance of this process [13]. Therefore, it is necessary to establish an accurate and reliable underwater laser echo solution process. To solve this problem, many scholars have conducted extensive research on echo calculations, and have proposed various waveform decomposition methods. Three echo-decomposition methods are commonly used: waveform decomposition, deconvolution, and peak detection. The underwater echo is different to land echoes, because the water absorbs the laser, and the underwater environment is complex; therefore, these methods are not all suitable as an underwater echo solution. Deconvolution requires multiple fittings, and is susceptible to noise interference and overfitting, resulting in a lack of precision and accuracy; peak detection lacks global consideration, leading to poor local accuracy. Because the Gaussian function is similar to the laser pulses emitted by the Lidar, and the complex echo signal can be fitted by the Gaussian function, waveform decomposition is a more suitable method for underwater echo solutions [14,15,16]. Hofton proposed that the full waveform is composed of several Gaussian components, and used the waveform’s initial half-width and amplitude estimates to decomposed it [17]. Chauve reported that it is appropriate to decompose the echo into a mixture of Gaussian components, and introduced more models into the waveform process than the Gaussian model [18]. Gaussian decomposition involves (1) Gaussian parameter estimation, and (2) Gaussian parameter optimization.

There are various optimization methods for these parameters: genetic, simulated annealing, particle swarm optimization (PSO), Levenberg–Marquardt (LM), etc. [19]. PSO is widely used in the optimization of laser-echo parameters. Li applied PSO to parameter optimization, and achieved good results [20]. Oscillating particle swarm optimization has been improved on the basis of PSO, and has also been applied to underwater echo solutions in recent years. Lu et al. applied two-order oscillating particle swarm optimization (OPSO) to underwater measurement, which improved the problem of the calculation process easily falling into local optimum and slowing convergence [21]. LM has also been combined with other methods for parameter optimization [22,23]. However, the above waveform decomposition methods are not effective in superimposed or weak waves; PSO focuses on the global characteristics, and ignores the layout characteristics. Although OPSO has solved the problem of easily falling into a local solution, it also sacrifices some optimization accuracy. LM has high optimization accuracy, but it is more dependent on the initial value, meaning that these methods cannot be directly applied to underwater echo solutions.

Therefore, this study used a layer-by-layer stripping inflection point decomposition method to obtain the Gaussian estimation parameters of the underwater echo, and then used OPSO and fixed damping LM to fit the estimated parameters, as some scholars have applied the combination PSO and LM to other fields and achieved good results [24]. In order to verify the effectiveness of this method, this study used a drone equipped with a 532 nm laser to detect a pond as the study background, collected relevant data, tested multiple experimental data using the process in this paper, and compared them with traditional algorithms, to verify the effectiveness of the underwater echo calculation process.

## 2. Algorithm Principle

### 2.1. Principle of the Gaussian Decomposition Method

The Gaussian decomposition method is based on the assumption that the laser-echo signal is constructed by multiple Gaussian functions; nowadays, this method has also been confirmed by many scholars, and has been applied to waveform calculation [25,26]. The purpose of Gaussian decomposition is to extract as many Gaussian components as possible, and obtain the information of each component (f(x), Equation (1)).
(1)$$f(x)=\sum_{i=0}^{n}A_{i}\exp(-\frac{{\left(x-u_{i}\right)}^{2}}{2\delta_{i}{}^{2}})+noise$$

f(x) is the amplitude value of the waveform data, *x* is the sampling time of the waveform data$;\;A_{i}$, $u_{i}$, $\delta_{i}$ are the peak value, center position, and half width of the Gaussian component respectively; noise is the background noise of the signal.

### 2.2. Data Preprocessing

Preprocessing includes noise estimation and the Savitzky–Golay smoothing filter; because the waveform decomposition process is easily affected by noise, it is necessary to preprocess the noise before waveform processing. Firstly, noise is estimated, using the Savitzky–Golay method to remove noise, and then using the five-point three-time smoothing method to smooth the waveform, in order to avoid generating false inflection points that would result in errors in the Gaussian decomposition [16,27]. The fluctuation of the waveform at the beginning and end stages is mainly due to the influence of noise; therefore, this study used the 5% waveform intensity at the beginning and end stages as the estimated noise, and then used the Savitzky–Golay and five-point three-time smoothing methods to filter out the noise, and smooth the waveform curve after filtering. It is necessary to set a threshold to avoid a large number of pseudo-waveform components (Figure 1); if the waveform intensity is below the threshold, there is no need to add new Gaussian components to fit the waveform. In this process, the threshold ($\sigma_{th}$, Equation (2)) is affected by the estimated noise and root-mean-square error before and after filtering ($\sigma_{n}$, Equation (3)).
(2)$$\sigma_{th}=3\sigma_{n}+noise$$
(3)$$\sigma_{n}=\sqrt{\frac{\sum_{1}^{n}{\left(y\left(i\right)-fx\left(i\right)\right)}^{2}}{n}}$$

$\sigma_{n}$ is the root-mean-square error before and after filtering; y(i), $f_{x}\left(i\right)$ are the signal strengths before and after filtering; *n* is the number of sampling points.

### 2.3. Improved Gaussian Decomposition Method

This study used the iterative decomposition method to find the maximum value of the waveform ($f_{m}$, Equation (4)), if the value satisfied the threshold condition, starting the waveform decomposition. Before each waveform decomposition, it is necessary to determine the number of inflection points in the waveform, and mark them, to determine the standard deviation of the waveform. This study used the ‘Quick Locating Algorithm for Turning Points in Discrete Point Set of Curve’ to detect waveform inflection points (*P*, Equations (5) and (6)) [28]. When all inflection points in this cycle were found, we needed to classify these points and count their number ($R\inf_{r}$, $L\inf_{l}$, Equation (7)), and we then established a calculation method for different standard deviations, according to the classification of left and right inflection points. If $R\inf_{r}$ and $L\inf_{l}$ are equal to 1, it indicates that there is no overlapping wave at the peak, but it does not mean that the standard deviation is not affected by other waveforms, so the standard deviation needs to be judged by Rinf and Linf (δ, Equation (8)). If there are other cases, it can be considered that the waveform has a superposition, or that the inflection point is not judged, owing to the large amplitude (δ, Equation (9)). After completing the above steps, the iterative solution is repeated, until all the peaks are lower than the threshold value, or reach the upper limit of the iteration.
(4)$$f_{m}=\max(f_{x}(i))>a$$

$f_{x}\left(i\right)$ is the waveform intensity at each time, *a* is the threshold value.
(5)$$S_{i-1,i}(t_{i+1},y_{i+1})=(t_{i}-t_{i-1})(y_{i+1}-y_{i-1})+(y_{i-1}-y_{i})(t_{i+1}-t_{i-1})$$
(6)$$\begin{array}{cc} P=(x_{i},y_{i}) & S_{i-1,i}S_{i,i+1}<0 \end{array}$$

$S_{i-1,i}$ is an evaluation index used to judge whether it is an inflection point; if $S_{i-1,\;i}$, $S_{i,i+1}<0$, the inflection point is *P*.
(7)$$R\inf_{r}=R\inf-R\inf_{j};L\inf_{l}=L\inf-L\inf_{j}$$

$R\inf_{r}$,$\;L\inf_{l}$ are the number of inflection points on the right and left sides of the peak; Rinf, Linf are the number of all inflection points on the right and left side of the peak; $R\inf_{j}$ is the number of inflection points at the right junction of the peak$;\;L\inf_{j}$ is the number of inflection points at the left junction of the peak.
(8)$$\begin{array}{c} L\inf_{l}=1\\ R\inf_{r}=1 \end{array}\{\begin{array}{ccc} \delta =medium(\left|\mu -P_{xi}\right|,\frac{\left|\mu -tgl\right|}{\sqrt{2\ln2}}) & , & L\inf=1\&R\inf>1\\ \delta =medium(\left|\mu -P_{xi}\right|,\frac{\left|\mu -tgr\right|}{\sqrt{2\ln2}}) & , & R\inf=1\&L\inf>1\\ \delta =medium(\left|\mu -P_{xi}\right|,\frac{\left|\mu -dis\right|}{\sqrt{2\ln2}}) & , & Others \end{array}$$

μ is the time of the peak value; $P_{xi}$ is the time of the inflection points on the left and right sides of the peak.

(1)When *L*inf = 1 and *R*inf > 1 (Figure 2a), the left-hand side of the peak is closer to the real situation. Therefore, taking the half position of the left peak as the full width at half maxima (*tgl*), and finding the closest point of the half peak to calculate the standard deviation and, finally, comparing these values, allows us to obtain the standard deviation of the peak (δ).(2)When *R*inf = 1 and *L*inf > 1 (Figure 2b), the right-hand side of the peak is closer to the real situation. Therefore, taking the half position of the right peak as the full width at half maxima (*tgr*), and finding the closest point of the half peak to calculate the standard deviation and, finally, comparing these values, allows us to obtain the standard deviation of the peak (δ).(3)When *R*inf = 1 and *L*inf = 1 (Figure 2c) or *R*inf > 1 and *L*inf > 1 (Figure 2d), the peak is a single waveform, or wrapped by multiple waveforms without superposition. Therefore, it is necessary to calculate the distance from the points closet to the half width peak on both sides to the peak point. The point with a smaller distance is taken as the full width at half maxima (*dis*), and these values are compared, to obtain the standard deviation of the peak (δ).
(9)$$Others\{\begin{array}{ccc} \delta =\min(\left|\mu -P_{xl}\right|,\frac{\left|\mu -tgl\right|}{\sqrt{2\ln2}}) & , & L\inf=1\&R\inf>1\\ \delta =\min(\left|\mu -P_{xl}\right|,\frac{\left|\mu -tgr\right|}{\sqrt{2\ln2}}) & , & R\inf=1\&L\inf>1\\ \delta =\frac{\left|\mu -dis\right|}{\sqrt{2\ln2}} & , & R\inf>1\&L\inf>1\\ \delta =\frac{\left|\mu -dis\right|}{\sqrt{2\ln2}} & , & Others \end{array}$$

(1)When *L*inf = 1 and *R*inf > 1 (Figure 3a), the left side of the peak is closer to the real situation; the right side of the peak exhibits a waveform superposition. Therefore, we need to find the corresponding inflection point, and the point of the nearest half peak position on the left side (*tgl*), and calculate their respective standard deviation (δ). This paper selected a smaller value as the standard deviation, to avoid error due to unrecognized partial inflection points.(2)When *R*inf = 1 and *L*inf > 1 (Figure 3b), the right side of the peak is closer to the real situation; the left side of the peak exhibits a waveform superposition. Therefore, we need to find the corresponding inflection point, and the point of the nearest half peak position on the right side (*tgr*), and calculate their respective standard deviation (δ). This paper selected a smaller value as the standard deviation, to avoid error due to unrecognized partial inflection points.(3)There are three different situations when *R*inf > 1 and *L*inf > 1 (Figure 3c–e); at this time, the distance from the center to the inflection points cannot accurately express the standard deviation, because the inflection points are affected by the superimposed waveform. In this study, the average position of the inflection points on the left and right sides of the peak are denoted as S1 and S2, respectively. The distance from the closest half position of the peak to the peak, on both sides of the peak, is calculated, and the shortest distance is denoted as S3. We sorted S1, S2, and S3, and obtained the middle distance as the half peak width (*dis*) to calculate the standard deviation (δ).(4)In this case, because the inflection point is not judged (Figure 3f), this study found the point closest to the half width on both sides of the peak, and calculated the distance between it and the peak, using the shortest distance as *dis*, to calculate the standard deviation (δ).

### 2.4. Two-Order Oscillating Particle Swarm Optimization

Two-order OPSO is a method based on PSO [29,30]; the basic idea is as follows:(1)According to the estimated parameters used to establish a group of particles, these particles are randomly distributed within the estimated parameter range.(2)The fitness of each particle (*fitness*, Equation (10)); if the fitness is better than the individual extremum, we update the individual extremum; if the fitness is better than the global extremum, we update the global extremum.(3)Optimizing the particles’ position and velocity ($V_{i}$, $X_{i}$, Equation (11)).(4)We repeat the above steps until the limit error requirement, or the number of iterations, is reached.
(10)$$fitness=\sum_{i=1}^{n}{\left(y_{i}-fx_{i}\right)}^{2};\;fx_{i}=\sum_{i=0}^{m}A_{i}\exp(-\frac{{\left(x-u_{i}\right)}^{2}}{2\delta_{i}{}^{2}})+noise)$$$y_{i}$ is the objective function; $fx_{i}$ is a function composed of multiple Gaussian components after Gaussian decomposition; *n* is the number of samples.
(11)$$\begin{array}{c} V_{i}(t+1)=wV_{i}(t)+c_{1}r_{1}(P_{i}-1+\alpha_{1})X_{i}(t)-\alpha_{1}X_{i}(t-1))\\ +c_{2}r_{2}(P_{g}-(1+\alpha_{2})X_{i}(t)+\alpha_{2}X_{i}(t-1))\\ X_{i}(t+1)=X_{i}+V_{i}(t+1) \end{array}$$
$X_{i}\left(t-1\right)$ is the position of the last time$;\;X_{i}\left(t\right)$*,*
$X_{i}\left(t+1\right)$ are the particle positions before and after updating; $V_{i}\left(t+1\right)$ and $V_{i}\left(t\right)$ are the particle velocity before and after updating; ω is the inertia weight factor; $\alpha_{1}$ and $\alpha_{2}$ are parameters to adjust the global search ability and local search ability of the algorithm$;\;c_{1}$ and $\;c_{2}$ are study factors$;\;r_{1}$*,*
$r_{2}$ are random numbers between 0 and 1,$\;P_{i}$ and $P_{g}$ represent the individual extremum and global extremum, respectively.

### 2.5. Levenberg–Marquardt Algorithm

The LM algorithm is a commonly used nonlinear least-squares method [31], which can optimize the estimated parameters, to make the estimated curve closer to the original curve ($\Delta x_{lm}$, Equation (12)). This paper used the gain ratio (ρ, Equation (13)) to adjust the damping coefficient (α, Equation (14)), which not only ensures that each iteration decreases in the direction of the objective function value, but also converges rapidly near the optimal value.
(12)$$\Delta x_{lm}=p-p^{\left(0\right)}={\left[J{\left(x\right)}^{T}J(x)+\alpha E\right]}^{-1}J{\left(x\right)}^{T}[y-f(x_{i},p^{\left(0\right)})]$$

$f\left(x_{i},\;p^{\left(0\right)}\right)$ is a function composed of undetermined coefficients; $p^{\left(0\right)}$ is an undetermined coefficient; J(x), $J{\left(x\right)}^{T}$ are the Jacobian matrix and transposed Jacobian matrix of $f\left(x_{i},p^{\left(0\right)}\right)$; y is the objective function, α is the damping coefficient.
(13)$$\rho =\frac{F(x)-F(x+h_{lm})}{L(0)-L(h_{lm})}$$

$F\left(x\right)-F\left(x+h_{lm}\right)$ is the residual change,$\;L\left(0\right)-L\left(h_{lm}\right)$ is the parameter change.
(14)$$\left\{ \begin{array}{l} \rho >0,\alpha =\alpha *\max(\frac{1}{3},1-{\left(2\rho -1\right)}^{3}),v=2\\ else,\alpha =\alpha *v,v=2*v \end{array}\right.$$

v is a parameter with an initial value of 2; if ρ>0, we reduce the value of α and update all parameters; otherwise, we expand α and update the parameters. The purpose is to adjust the step α by adjusting the change coefficient v; when the drop is too slow, we use a larger α to make the method close to the gradient method. When the drop is too fast, we use a smaller α to bring the method close to the Gauss–Newton method.

## 3. Results

### 3.1. Parameters Estimation

This study used a drone equipped with a 532 nm laser to detect a pond as the study background; the pond is located in a park in Guilin, Guangxi, China. The water quality is clear and the water depth is shallow, and the interference to the lidar signal is not strong. Through a preliminary analysis of the echo signal, we obtained the corresponding waveform intensity of each experiment, as the experimental data. Firstly, the original data were preprocessed to remove the noise in the waveform, and the data were smoothed by the Savitzky–Golay and five-point three-time smoothing methods, then we calculated the root-mean-square for before and after filtering, and the noise. The waveform threshold was estimated according to the root-mean-square, in order to provide a threshold for whether to continue to decompose the waveform later (Figure 1). After we obtained the preprocessed waveform signal, the Gaussian decomposition method was used to decompose the waveform signal into multiple Gaussian function components. In the first decomposition process (Figure 4a), $R\inf_{r\;}=1$, $L\inf_{l}=1$, we needed to confirm the number of inflection points on both sides of the peak, Rinf =1, Linf >1; therefore, the left side of the peak is closer to the real situation. We need to extract the inflection points on both sides, and the left side closest to the half peak point, for calculation to get σ. In the second decomposition process (Figure 4b), the peak is surrounded by multiple waveforms, Rinf >1, Linf >1, so we need to extract the inflection points on both sides, and the points closest to the half peak on both sides for calculation to get σ. In the third decomposition process (Figure 4c), $R\inf_{r\;}=1$, $L\inf_{l}>1$, we found that there was waveform superposition on the left side of the peak, the waveform on the right side was closer to the real situation; we extracted the inflection point on the right side and the point closest to the half peak on the right side for calculation to get σ. In the fourth decomposition process (Figure 4d), there was no inflection point on one side of the waveform; the inflection point could not be used to decompose the waveform. It was necessary to extract the point on both sides of the peak closest to the half peak, and take the point with the shortest distance to the peak for calculation to get σ. In the fifth decomposition process (Figure 4e), the situation is similar to the fourth decomposition, so we used the same method of calculation to get σ.

### 3.2. Parameter Optimizing and Fitting

After the initial decomposition of the parameters, some meaningless Gaussian components will be generated. These meaningless Gaussian components will produce some pseudo waveforms, and affect the waveform fitting (Figure 5a). Therefore, we need to eliminate these Gaussian components during the iteration (Figure 5b), and if any Gaussian components are less than one emission-pulse in width after decomposition, we can eliminate the component with a smaller wave peak [32]. The parameters are estimated in the initial decomposition process, which cannot represent the real value; negative oscillation occurs when compared with the original data (Figure 6), and it is necessary to optimize and fit the estimated parameters. Optimizing the characteristic parameters can achieve better results before parameter fitting [33]. The underwater waveform strength of the research data is very low, because water will absorb most of the laser. Although PSO (Figure 7a) can achieve better results in error calculation, its global performance is worse than that of OPSO (Figure 7b), and easy to fall into local optimum, so this method cannot deal well with a weak wave. This paper used OPSO to optimize the characteristic parameters. We found that PSO had a good fitting effect on the Gaussian component with a large peak value, but the fitting accuracy for weak waves was far less than that obtained by using OPSO; further parameter optimization fitting is needed, after completing the optimization of the characteristic parameters. The LM algorithm is easily affected by the initial value, and the peak value in the previous Gaussian decomposition process adopts the estimated value, and has a large deviation from the original waveform, which will affect the optimization accuracy. This is the reason why OPSO was used to optimize the parameters in the previous process. A large number of scholars have proposed that the fixed damping coefficient can help in the calculation, but this method was not applicable in this study (Figure 8a), because the fixed damping coefficient is easy to fall into the local optimal solution, and cannot express the original waveform data well. Compared with the fixed damping coefficient, the LM algorithm using the variable damping coefficient has a better fitting effect, and better globality (Figure 8b).

### 3.3. Accuracy Comparison

To verify the accuracy of the decomposition method, four groups of representative underwater echo data were selected, and these waveforms were decomposed using different Gaussian decomposition methods. The distance from the left and right inflection points of test 1 and test 2 to the peak was different; the boundary of the strong and weak waveform signal of test 1 was not obvious, which had a superposition phenomenon. The boundary of the strong and weak waveform signal of test 2 was obvious. The distances from the left and right inflection points of test 3 and test 4 to the peak were the same; the boundary of the strong and weak waveform signal of test 3 was not obvious; and the boundary of the strong and weak waveform signal of test 4 was obvious.

Through comparison, it was found that although the number of waveform components is three, after removing the pseudo-Gaussian component, the method proposed in this paper had a better waveform-fitting effect, and was closer to the original waveform. After comparing a variety of data, it was found that the decomposition effect of the inflection point method was poor; when the distance between the inflection point and the peak value was not equal (Figure 9), the peak dislocation phenomenon occurred. Moreover, the Gaussian component with a smaller peak value did not fit well with the original data, meaning that the inflection points method alone could not correctly express the original waveform. When using the iterative method to process the waveform (Figure 10), it was also found that the method had a poor effect on the waveform decomposition with smaller peaks, and the final decomposition result could not fully express the original waveform; when using the inflection point selection decomposition method (Figure 11), it was found that the fitting situation with a smaller peak value was improved, compared with the first two methods, but the fitting effect at the junction of two Gaussian components was not good. The main reason for this is that it is unreasonable to calculate δ by taking the minimum value of the half-peak width as the standard deviation, when the inflection points on both sides of the peak are 1. This could easily cause the situation of test 3; the waveform was not completely decomposed and appeared as multiple pseudo components because of the small δ. It would produce large errors after the subsequent removal of the pseudo components, and had great trouble with the subsequent parameter optimization fitting. When using the method presented in this paper (Figure 12), we found that the method in this study obtained good conditions for each group of data, however, compared with the previous situation. It has been well improved, in addition to the poor fitting condition of test 4 at the peak junction but, compared with other methods, a similar problem also occurs, except for the iterative method. Although the iterative method has a good fitting effect at the junction, it causes a large error in the smaller peak, in order to fit the junction. The poor fitting of the junction is mainly due to the use of the local maximum value to represent the peak value of the Gaussian component, but it is obviously unreasonable to use the maximum value in the superimposed waveforms, which is also the focus of our planned follow-up research.

To more clearly see the decomposition accuracy of each method for the experimental data, this study made an accuracy verification comparison (Table 1), and calculated the root-mean-square error (*RMSE*, Equation (15)) and the coefficient of determination (*R*^2^, Equation (16)). The results showed that the method in this paper is more accurate than other methods; although the accuracy of some data is not higher than in the iterative method, the accuracy of the iterative method is not much higher than that of this paper, and it can be found that the overall fitting effect of this paper is better than the iterative method. The results showed that the *RMSE* is 2.544 and the *R*^2^ is 0.995975, the average precision parameters are higher than other data, and the decomposition effect is more stable and accurate. In summary, this method is more suitable for the decomposition of multiple waveforms; not only that, but this method can also express waveform data more accurately than the other methods.
(15)$$RMSE=\sqrt{\frac{1}{N}\sum_{i=1}^{n}{\left(y_{i}-f(x_{i})\right)}^{2}}$$

*N* is the number of sampling points; $y_{i}$, $f\left(x_{i}\right)$ are the signal echo intensities before and after optimization, respectively.
(16)$$R^{2}=1-\frac{\sum_{i=1}^{n}{\left(y_{i}-f(x_{i})\right)}^{2}}{\sum_{i=1}^{n}{\left(y_{i}-\bar{y}\right)}^{2}}$$

$\bar{y}$ is the mean value of each signal, and an *R*^2^ value closer to 1 indicates a better fit.

In order to verify the effectiveness of this method in complex waveforms, and the limitations of other algorithms in decomposition, a complex waveform superimposed by multiple Gaussian functions is constructed for simulation experiments (Figure 13), and solved using the inflection point method (Figure 14a), the iterative method (Figure 14b), the inflection point selection decomposition method (Figure 14c), and the proposed method (Figure 14d). Its parameters are set from left to right to generate a complex waveform (Table 2). After the preliminary decomposition of the waveform, it can be found that after the inflection point method, the waveform has a large number of waveform components, because the superimposed wave will produce some pseudo inflection points, which will affect the judgment of the decomposition process of the waveform component. Therefore, the inflection point method cannot have the condition for complex waveform decomposition, and when the waveform cannot be judged by multiple superposition of the half peak width, iterative decomposition will yield a large δ. Therefore, the iterative method cannot be applied to complex waveform decomposition. The inflection point selection method is improved on the basis of the iterative method, but it is unreasonable to select the shortest half peak width to calculate δ when the inflection points on both left and right sides are 1; moreover, using the iterative method, when multiple waveforms are superimposed but the number of left and right inflection points is 1, excessive δ will cause the incorrect decomposition of waveforms. Although this method can roughly judge the center position of the original waveform after eliminating the spurious variables, the peak value and δ that deviate too much from the original waveform will increase the workload of subsequent parameter optimization. When using the method in this paper, all the waveform components were roughly judged, and the parameters were not much different from the original data. This method can provide a better estimation parameter for parameter optimization. After obtaining the estimated parameters, using the OPSO and LM algorithms to optimize the parameters (Figure 15), and comparing the optimized waveform with the original waveform (Table 3), the *RMSE* was only 0.0016,the *R*^2^ was 1, and the optimized waveform was very close to the original waveform.

In order to further verify the correctness of the algorithm flow in this study, this study selected the worst accuracy among the four tests (test 4) and used this test’s data as study data, using a variety of optimization methods to optimize the estimated parameters of these data (Table 4). The results show that using PSO and updating the LM algorithm had the highest accuracy, but from the fitting effect of the final results (Figure 16), this method did not achieve a good fitting effect, it fell into the local optimal solution in order to pursue accuracy, and it can be seen that the fitting effect of weak waves is not good except when using OPSO or combining OPSO and update damping LM. It can also be found that the optimization method combining the two methods can improve the waveform fitting accuracy, and is closer to the original waveform curve.

## 4. Discussion

The laser radar waveform solution is the initial process for obtaining the point cloud data, and the accuracy of the solution directly affects the quality of the subsequent point cloud. This paper proposes an improved inflection point selection decomposition method, and establishes a set of corresponding laser radar solution processes, which has been greatly improved in accuracy and stability, compared with other decomposition methods.

The reason why the decomposition accuracy of this study is more accurate than other methods is that the corresponding waveform parameter solution method is established by each different waveform feature, and this method can use the most suitable calculation model to decompose the waveform. Compared with other decomposition methods, this study combines the advantages of the inflection point method and the iterative method, to analyze the waveform data layer by layer. According to the position of the inflection point and the half-peak width to establish the best decomposition method, when the inflection on both sides of the decomposed waveform is 1, the waveform may not be a superimposed waveform, by calculating the distance from the inflection points on both sides to the peak center, and the distance from the half-peak width points on both sides to the peak center. Then, the maximum and minimum values are removed to obtain δ. In this way, if there are two same values in the judgment, the same value will be reserved. When the inflection point of the waveform on both sides is greater than 1, this waveform may be superimposed, calculating the δ by the distance from the center point of the two inflection points closest to the center on both sides to the peak value, and the distance from the half-peak width of the closest peak to the peak. The purpose of selecting the center of the two inflection points is that the median value of the two inflection points does not affect the decomposition of the superimposed wave, and there will be no superimposed wave that is decomposed as a single wave because of the large δ, and no multiple pseudo components with small δ. When the inflection point on one side of the waveform is 1, it shows that the side is closer to the waveform to be decomposed, and it is more reasonable to calculate the δ on this side. When the inflection point on both sides of the waveform is 0, it is considered that the inflection point is not calculated because the corresponding distance of the amplitude of the waveform component is too close, the half-peak width needs to be used to calculate δ. In the process of parameter optimization, although the accuracy and solution efficiency of PSO and fixed damping LM may be higher than the method in this paper in the face of the waveform component, the peak difference is not large; when facing an underwater waveform with weak waves, it easily falls into the local optimal solution. In order to verify the efficiency of the method, we calculated the speed of waveform calculation. We used the Intel Core i3-8350K @ 4.00 GHz quad-core processor, 32 GB (Kingston DDR4 3200 MHZ 32 GB), NVIDIA Quadro P1000 graphics card, and Asus PRIME-Z370-A (Z370 chipset) motherboard to configure the computer to run the waveform calculation program. The final solution time was 0.000156 s. Therefore, this paper adopts a more global parameter optimization method that can better meet the solution requirements of underwater waveforms.

The method in this paper also has certain limitations:(1)When solving the superimposed waveform, it is unreasonable to use the highest peak value as the amplitude of the Gaussian component; although this limitation can be corrected in the subsequent parameter optimization process, it undoubtedly does not increase the workload of parameter optimization.(2)The laser echo signal will be affected by noise in the water. In this paper, the water data under the influence of different noise are not further explored, but this problem will be further studied in follow-up work.(3)Although this method uses multiple sets of research data, the experimental environment is still limited to ponds with clear water quality. Because the data source problem can only be tested with simulated data, there is no exploration of different types of water body data, and the universality of this method cannot be verified.

## 5. Conclusions

This paper proposes an improved waveform decomposition method based on inflection point selection decomposition, and we obtained the following conclusions:(1)By comparing multiple sets of study data, underwater lasers are mostly complex waveforms of strong wave and weak wave superposition. We found that the improved inflection point selection method proposed in this paper is more suitable for underwater waveform calculation, which can solve underwater waveform data well and has a good effect on the parameter extraction of superimposed waves and weak waves.(2)To further verify the universality of this method, this study used a variety of methods to decompose the simulated waveform. The results showed that the waveform decomposed by this method is closer to the actual waveform in comparison with other methods, and the accuracy is much higher than that of other decomposition methods.(3)To verify the accuracy of the subsequent parameters optimization process, this study compared a variety of parameter optimization combinations, and the results showed that although the accuracy of the optimized method was not the highest, the actual results were closer to the original waveform than the other methods.

Our method highlighted the following aspects as worthy of future research and improvement:(1)The underwater laser echo signal is susceptible to various noise sources. In follow-up work, multiple experiments will be carried out under the influence of various noise sources, to further verify the accuracy and reliability of the proposed method.(2)In follow-up work, a variety of different sensors will be used to test the algorithm in this paper, to further understand whether the limitations and deviations of the sensors will affect the results.(3)The data background selected in this paper is a pond in a park in Guilin, Guangxi, China; the water quality has little effect on the laser echo signal. The planned follow-up research will study different types of water bodies, such as oceans and rivers, to further verify and evaluate the universality of this method in different environments.

## Figures and Tables

**Figure 1 sensors-23-06058-f001:**
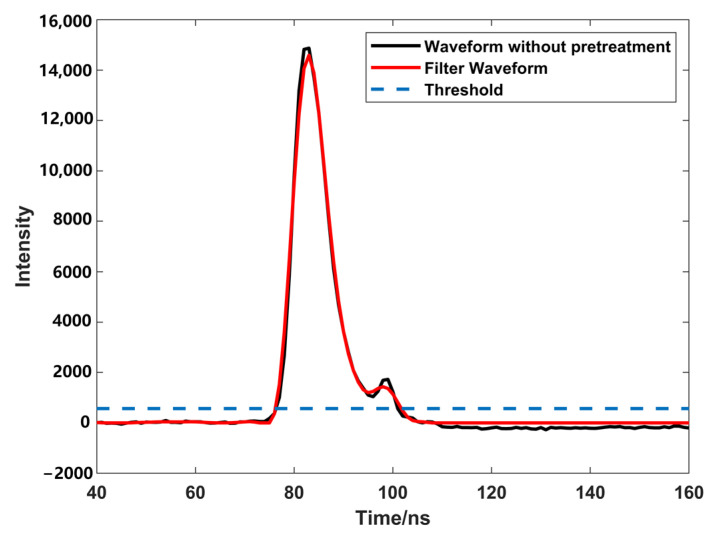
Waveform pre- and post-filtering.

**Figure 2 sensors-23-06058-f002:**
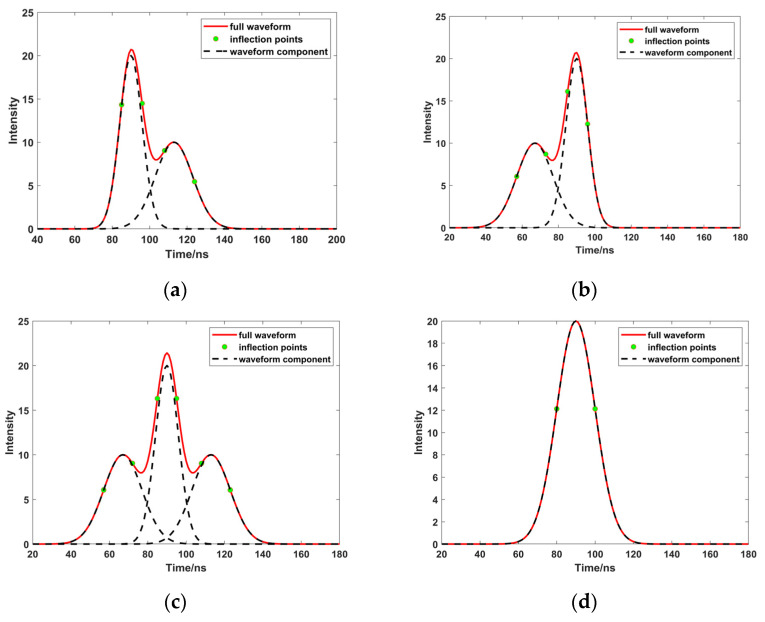
Different images of$\;L\inf_{l}=1$ and $R\inf_{r}=1\;$ ((**a**–**d**) represent different cases of $L\inf_{l}=1$ and $R\inf_{r}=1$, respectively).

**Figure 3 sensors-23-06058-f003:**
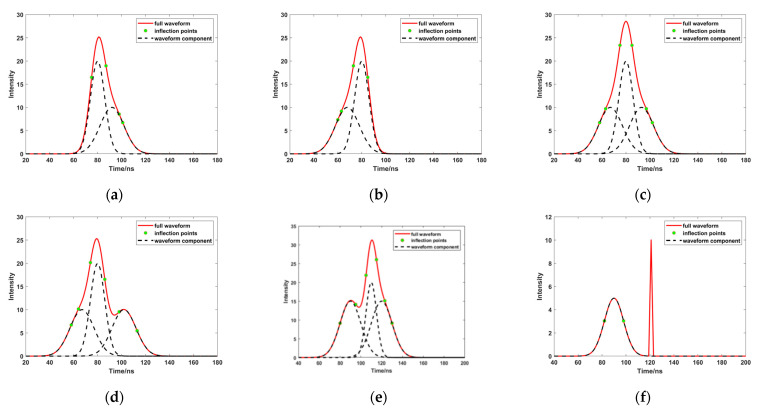
Different images of others ((**a**–**f**) represent different cases of others, respectively).

**Figure 4 sensors-23-06058-f004:**
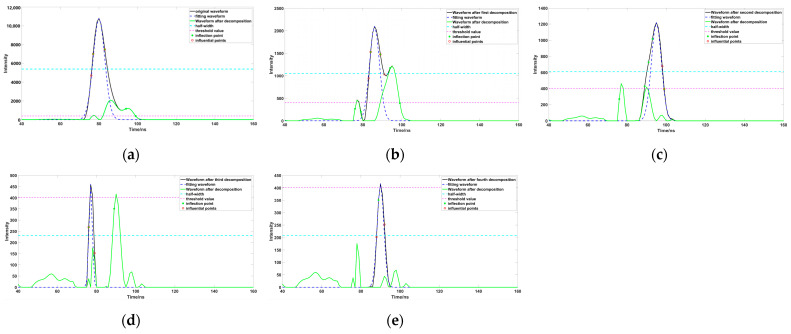
Waveform multiple decomposition process ((**a**–**e**) represent the diagrams of each stage in the process of waveform processing).

**Figure 5 sensors-23-06058-f005:**
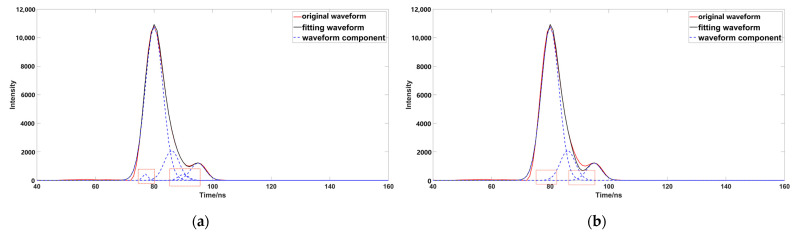
Comparison of Gaussian component elimination ((**a**,**b**) represent the image before and after the removal of the pseudo component, and the red-box-highlighted area is the difference between the two images).

**Figure 6 sensors-23-06058-f006:**
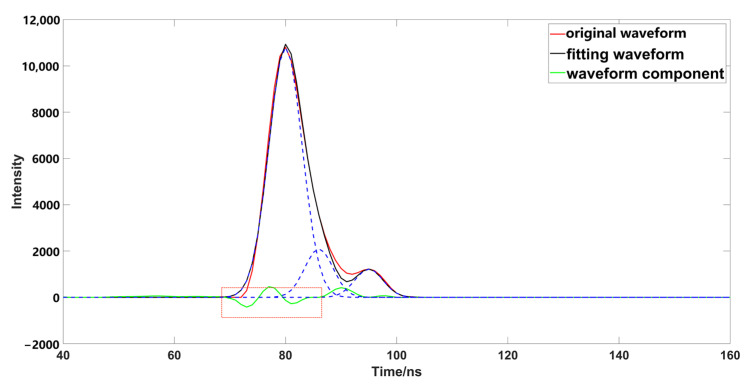
Negative oscillation occurs after decomposition (the red box highlights the negative oscillation area).

**Figure 7 sensors-23-06058-f007:**
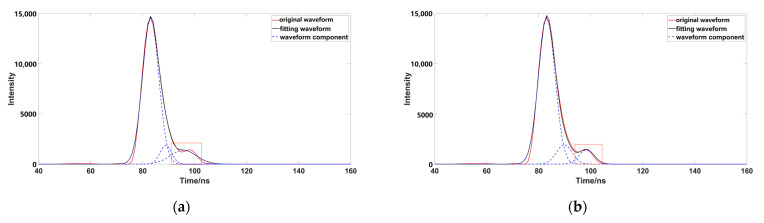
Comparison of different parameter optimization methods ((**a**,**b**) represent the waveform solution using PSO and OPSO, respectively; the red box highlights the area with a large gap between the different methods).

**Figure 8 sensors-23-06058-f008:**
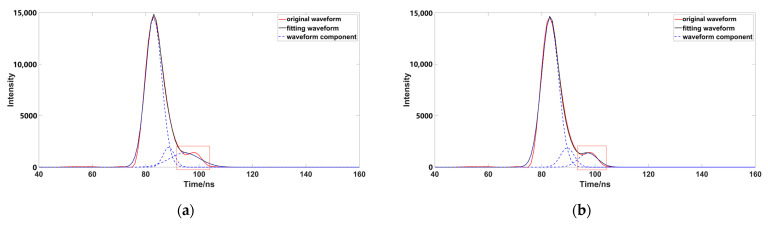
Comparison of fixed damping coefficient. ((**a**,**b**), respectively, represent whether to use fixed damping coefficient to solve the waveform, the red box highlights the area with the large difference between the different methods).

**Figure 9 sensors-23-06058-f009:**
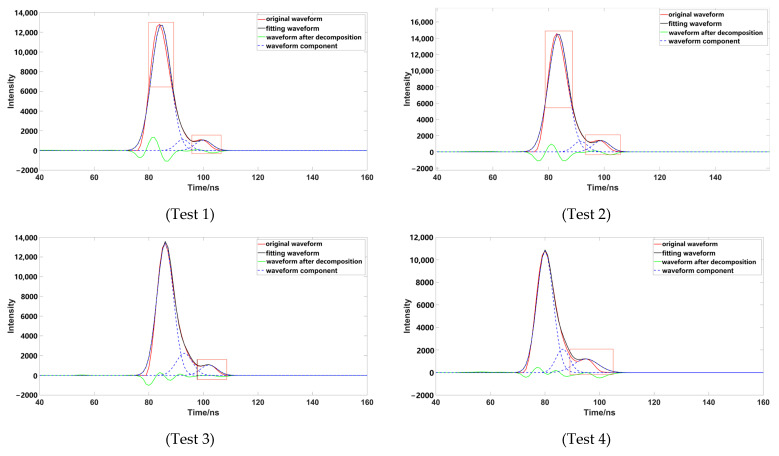
Data processed by inflection point (the image representation uses the inflection point to solve different data; the red box highlights the area that is poorly fitted).

**Figure 10 sensors-23-06058-f010:**
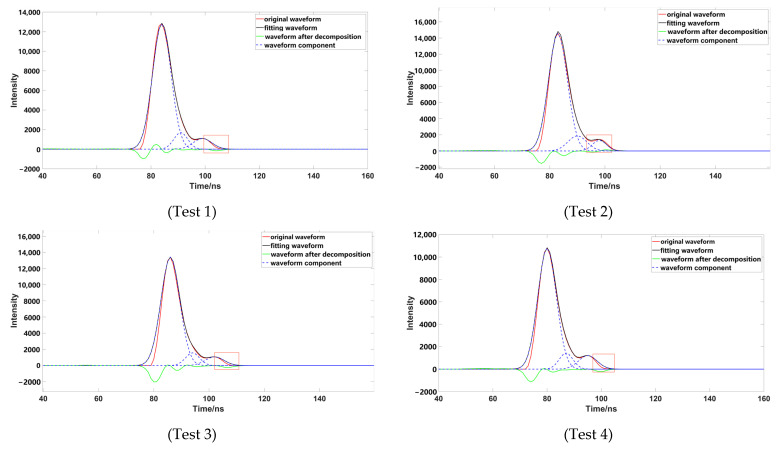
Data processed by iterative decomposition (the image representation uses iterative decomposition to solve different data; the red box highlights the area that is poorly fitted).

**Figure 11 sensors-23-06058-f011:**
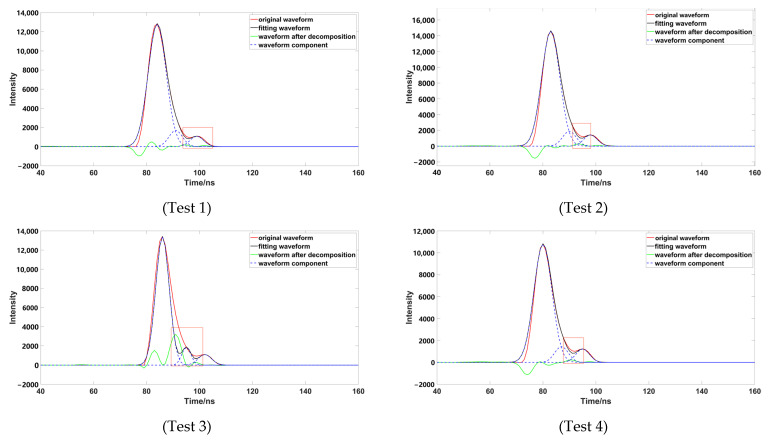
Data processed by inflection point selection decomposition (the image representation uses inflection point selection decomposition to solve different data; the red box highlights the area that is poorly fitted).

**Figure 12 sensors-23-06058-f012:**
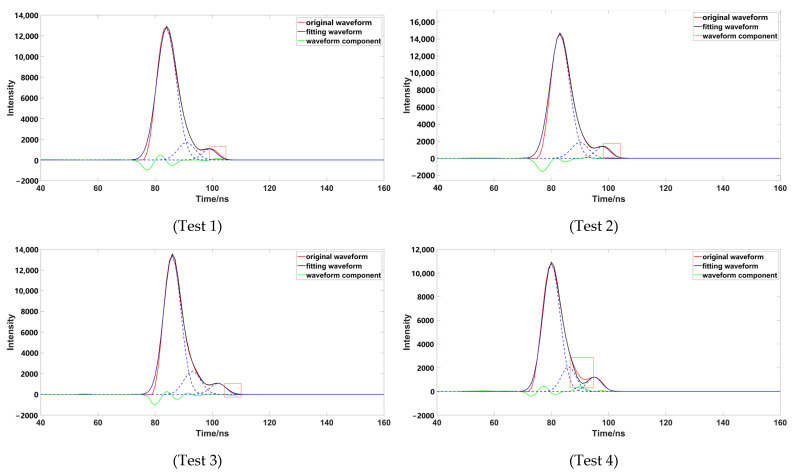
Data processed by this paper (the image representation uses the method of this paper to solve different data; the red box highlights the area that is poorly fitted).

**Figure 13 sensors-23-06058-f013:**
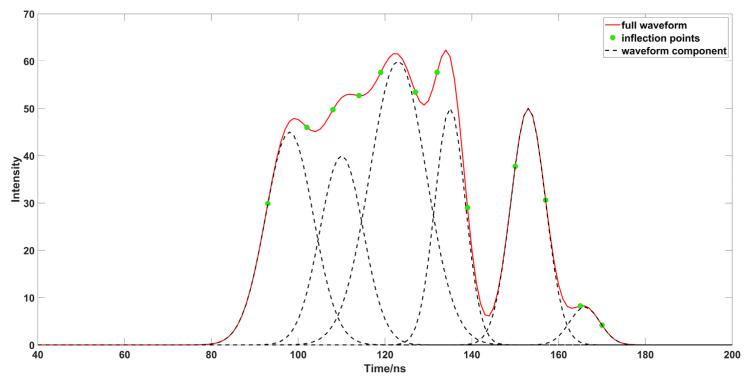
Waveform composed by multiple waveform components.

**Figure 14 sensors-23-06058-f014:**
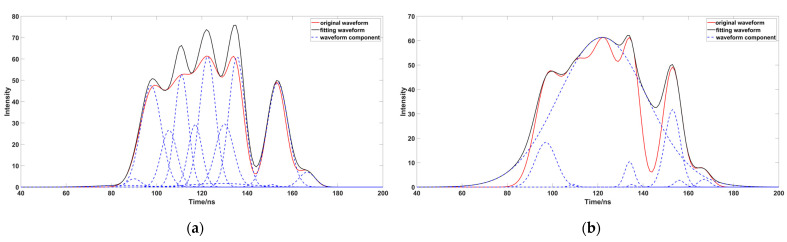
Different methods decompose the simulation waveform. ((**a**) is the result of using inflection point to solve the simulated waveform data; (**b**) is the result of using iterative to solve the simulated waveform data; (**c**) is the result of using inflection point selection decomposition to solve the simulated waveform data; (**d**) is the result of using the method in this paper to solve the simulated waveform data.)

**Figure 15 sensors-23-06058-f015:**
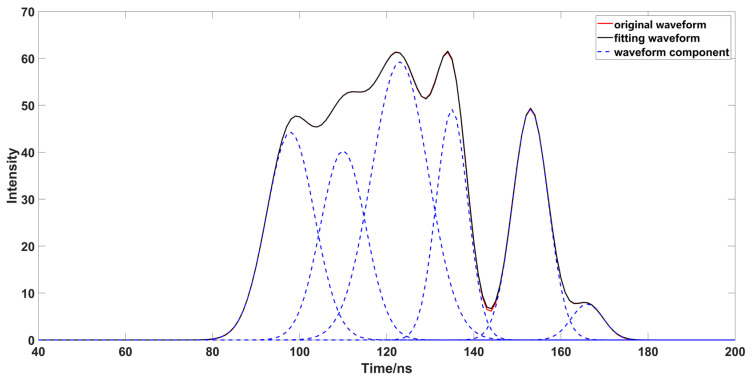
Comparison of original waveform and optimized waveform.

**Figure 16 sensors-23-06058-f016:**
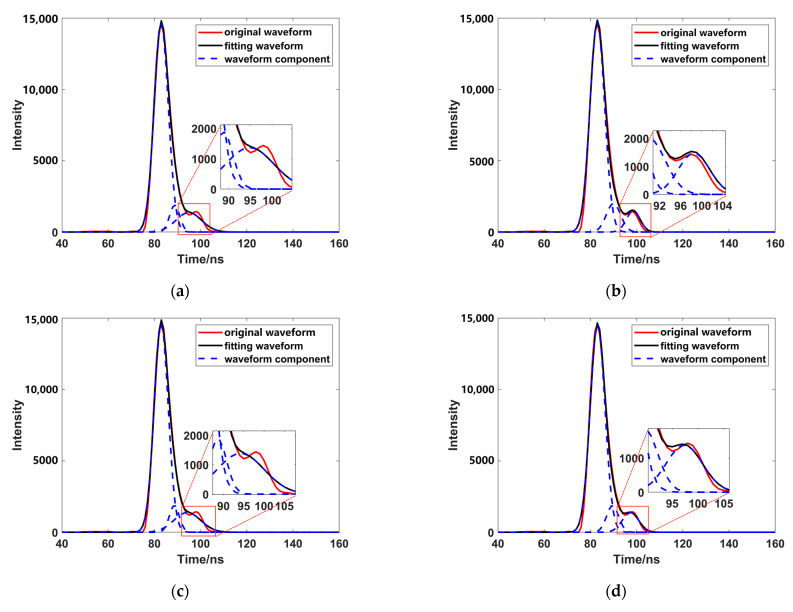
Fitting results of different optimization methods. ((**a**) is the result of using PSO to fit waveform data; (**b**) is the result of using OPSO to fit waveform data; (**c**) is the result of using fixed damping LM to fit waveform data; (**d**) is the result of using update damping LM to fit waveform data; (**e**) is the result of using PSO and fixed damping LM to fit waveform data; (**f**) is the result of using PSO and update damping LM to fit waveform data; (**g**) is the result of using OPSO and fixed damping LM to fit waveform data; (**h**) is the result of using OPSO and update damping LM to fit waveform data.)

**Table 1 sensors-23-06058-t001:** Decomposition accuracy of experimental data.

	Data	Iterative	Inflection Point	Inflection Point Selection Decomposition	This Paper
RMSE	Test 1	2.4600	4.1989	2.4647	2.6383
	Test 2	3.8405	4.1085	3.6739	3.7071
	Test 3	2.4133	2.4133	8.1702	2.3909
	Test 4	1.9612	1.9612	2.8015	1.4397
	average	2.66875	3.170475	4.277575	2.544
R^2^	Test 1	0.9964	0.9894	0.9963	0.9958
	Test 2	0.9929	0.9919	0.9935	0.9934
	Test 3	0.9964	0.9964	0.9591	0.9965
	Test 4	0.9966	0.9966	0.9931	0.9982
	average	0.995575	0.993575	0.9855	0.995975

**Table 2 sensors-23-06058-t002:** Characteristic parameters of simulated data.

i	1	2	3	4	5	6
A	45	40	60	50	50	8
μ	98	110	123	135	153	166
δ	5.5	5	6.5	3.5	4	3.5

**Table 3 sensors-23-06058-t003:** Characteristic parameters of simulated data decomposition.

	i	1	2	3	4	5	6
A	Actual value	45	40	60	50	50	8
	Optimized parameters	44.2589	40.3164	59.2484	49.1012	49.3643	7.6628
μ	Actual value	98	110	123	135	153	166
	Optimized parameters	97.8985	109.9738	123.0346	134.9677	153.0104	166.0331
δ	Actual value	5.5	5	6.5	3.5	4	3.5
	Optimized parameters	5.4893	5.1777	6.4799	3.6108	4.0575	3.6352
*RMSE*		0.0016					
*R^2^*		1					

**Table 4 sensors-23-06058-t004:** Comparison of different optimization methods.

Method	RMSE	R^2^
Original data	3.7071	0.9934
PSO (Figure 16a)	1.9853	0.9976
OPSO (Figure 16b)	2.2267	0.9970
Fixed damping LM (Figure 16c)	1.5947	0.9988
Update damping LM (Figure 16d)	1.9386	0.9982
PSO and fixed damping LM (Figure 16e)	1.6405	0.9987
PSO and update damping LM (Figrue 16f)	1.9485	0.9988
OPSO and fixed damping LM (Figure 16g)	1.5947	0.9988
OPSO and update damping LM (Figure 16h)	2.0387	0.9980

## Data Availability

Not applicable.

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
