# Peer review of "Method to Solve Underwater Laser Weak Waves and Superimposed Waves"

_sensors, 2023, doi:10.3390/s23136058_

Round 1

Reviewer 1 Report

1.     This paper presents a solution method for underwater laser weak wave and superimposed wave. The method provides satisfactory results and has good potential for applications in this field.

2.     The writing of the paper does not meet the publication standard and need significant improvement. Some sentences are too long and are difficult to read; for example, the last sentence in the Abstract is very long. There are also many grammatical errors throughout the paper. Some examples are given below.

3.     Line 43, “Underwater environment is usually have uneven…” delete “is” and change “have” to “has”

4.     Line 94, “…layout characteristics, Although OPSO solved…” change “,” to “.” or change Although to although.

5.     Lines 100-101, “there are some scholars have applied the combination PSO and LM to other fields”, delete “there are”

6.     Line 131, δn means sn?

7.     Line 153, “SI−1,I”, change I to i to be consistent with the equation.

8.     Lines 155 to 156, change “Rinfr , Linfl are the number of inflection points on the left and right sides of the 155 peak,” and “Rinf, Linf are the number of all inflection points on the left and right side of the peak,” to “on the right and left side”, not left and right side.

9.     Line 158, “at the lift junction of the peak”, change lift to left.

10. The quality of figures 2 – 12 is poor, which should be improved.

11. Equation 10, change fxi to fxi

12.  Equation 14, a = a*v, and v = 2*v are confusing. Suggest to rewrite them.

13.  Fig. 9, add the legend for the green line.

14.  It is recommended to introduce the general information of the pond and the process of the data collection by the drone. For example, what pond information was collected?

15.  It is suggested for the authors to carefully proofread the paper before the next submission.

The English language should be improved.

Author Response

Thank you very much for your affirmation of our article and algorithm. I am very sorry to trouble you with some grammatical problems in the writing process of the article. For the grammatical errors ( 2-5,8,9,11 ) you proposed in our article, we modified them one by one and marked the modified part with a yellow background.

6.We modify the wrong symbols and check other symbols of the article, we are very sorry to produce this low-level error.

10.We are also aware of the problem of low image resolution. We re-experiment the experimental data and obtain the relevant images, and replace the original low-resolution images. Thank you very much for raising our omission.

12.For the part of the formula that is easy to cause trouble, we reinterpret the formula in line 238 on page 7 of the article, and describe the two symbols in the formula in more detail. Among them, α is the gradient descent step size of the LM algorithm in the article. The main purpose of the description of v and α is to use a smaller α when the method drops too fast, so that the whole formula is close to the Gauss Newton method. Therefore, it is necessary to reset v to achieve the purpose of reducing the step size. When the decline is too slow, it is necessary to use a larger α to make the whole formula close to the gradient method, so the v * 2 in the formula is used to achieve the purpose of increasing the step size.

13.We are very sorry that we did not add the green line legend due to our negligence. Thank you very much for pointing out our negligence.

14.Your suggestion is very correct, and it is necessary to introduce the corresponding experimental environment. Therefore, we describe the experimental pond in the article. The pond obtained in this paper is a pond in a park in Guilin, Guangxi, China. The water quality is relatively clear and has a weak effect on laser intensity noise.

15.In the later article writing process, we will also pay attention to this problem, check the manuscript many times, and work hard on your valuable opinions on our article and your affirmation of us.

Reviewer 2 Report

This work focuses on processing underwater laser echo signals, which are challenging due to the presence of weak waves and superimposed waves. Existing methods for decomposing these waveform signals have limitations, leading to errors in data processing. To overcome these challenges, the study proposes an improved inflection point selection decomposition method. Using a drone equipped with a 532 nm laser, the authors collect data from a pond and compare their method with others. The proposed method shows greater stability and accuracy, with better parameter estimation. Additionally, the study optimizes the estimated parameters using OPSO and LM algorithms, resulting in waveforms that closely resemble the original signals. The method is further validated through the decomposition and optimization of simulated complex waveforms, demonstrating superior performance compared to other methods. 

Here some technical issues that could be improved in the work:

1.     Signal quality and noise: Underwater laser echo signals can be affected by various sources of noise, such as scattering, absorption, and ambient light interference. The accuracy and reliability of the proposed method may be compromised if the weak wave and superimposed wave signals are strongly influenced by noise. Handling and mitigating noise in underwater laser signals is a challenging technical problem.

2.     Sensor limitations: The effectiveness of the proposed method heavily relies on the capabilities and characteristics of the sensor used for data collection. The accuracy of the sensor in capturing underwater laser echo signals, its resolution, and its range limitations could impact the performance of the proposed method. It is crucial to understand the sensor's limitations and potential biases to ensure the reliability of the results.

3.     Algorithm complexity and computational resources: The proposed inflection point selection decomposition method, as well as the subsequent parameter estimation and optimization processes, may involve complex algorithms and computations. If the algorithms are computationally intensive or require significant computational resources, such as memory or processing power, it could limit the method's practical implementation and scalability.

4.     Generalization to different environments: The work mentions using a drone equipped with a laser to detect a pond as the study background. However, the performance of the proposed method may vary when applied to different underwater environments, such as oceans, rivers, or different types of water bodies. The generalization of the method and its robustness across diverse environments should be carefully evaluated.

5. The labels in all figures must be improved. Increase font size 

  1. Sentence structure and clarity: The sentences in the work are often long and contain multiple ideas, which can make the text difficult to follow. Some sentences lack clarity and conciseness, making it challenging to grasp the intended meaning immediately.

  2. Lack of transition and coherence: The work lacks explicit transitional phrases or words to connect ideas and create a coherent flow. This leads to a disjointed presentation of information and makes it harder for readers to understand the relationships between different sections.

  3. Use of technical jargon and terminology: The work frequently uses technical terms and concepts without providing sufficient explanations or definitions. This can be problematic for readers who are not familiar with the subject matter, hindering their understanding of the content.

Author Response

1.Thank you very much for some suggestions for improvement of our work. Noise is indeed a challenging technical problem in the application of underwater laser signals. Although the 532 nm laser used in the data in this paper has strong penetration in water, it is also easy to be affected by noise. In this paper, the noise is simply processed by simple S-G filter and five-point three-line method. Good experimental results can be obtained in clear ponds with weak noise influence. Due to the limitation of experimental data, this paper does not take into account the strong noise influence, which is also the deficiency of our experiment. We write this problem into the discussion and discuss it, which is located on line 467 on page 18 of the article. We take it as our next step, which is located in line 491 on page 19 of the article, and increase the diversity of data detection in the follow-up work to further verify the method in this paper.

2.Thank you very much for your valuable comments on us. Different sensors do have an impact on data collection. In this paper, 532 nm laser is used to detect the underwater environment mainly based on line 51 of page 2. Li found that 532 nm laser has strong penetrability to underwater laser, so other sensors are not considered to collect data for comparison. We will take this problem as the next step. The specific location is line 494 of page 19, which will be studied in the follow-up work.

3.The operation speed of the  algorithm is indeed an important aspect that needs to be considered in the waveform calculation problem. We also tested the algorithm for this problem. The algorithm running environment is : Intel Core i3-8350K @ 4.00GHz quad-core processor, 32GB ( Kingston DDR4 3200MHZ 32GB ), NVIDIA Quadro P1000 graphics card, ASUS PRIME-Z370-A ( Z370 chipset ) motherboard, in the calculation of running memory for the middle level. In this paper, the time of a waveform operation in the computer is 0.000156 seconds. We also explain this part of the content in the discussion. The specific location is the 455th line of the 18th page of the article.

4.Thank you very much for your suggestions on the future application of our method. The algorithm detection for different hydrological environments is the research direction of our research group 's follow-up plan. We also plan to test in different water areas near the coast to further verify the method of this paper. As our next step, the specific location is located on page 19, line 497. In the follow-up work, the water areas with different hydrological characteristics are tested to further verify the method of this paper.

5.We have improved the problem of image label fonts in the image, re-acquired the experimental data and replaced the image.

6.We are very sorry for the trouble caused by the English problems in this article. We also recheck and revise the sentences in the article. In the sentence structure, we also replace long sentences with multiple short sentences for understanding. In terms of transitional coherence, we will also recheck the article and strive to enhance the coherence and readability of the article. In the problem of too many technical terms, we also explain some of the concepts and terms mentioned in the abbreviation explanation at the end of the article. In the subsequent publication of the paper, we will also invite professional English polishers to correct the grammatical problems of the article.